# Measurements of Gaseous Hydrogen–Nitrogen Laser-Plasma

Christian G. Parigge 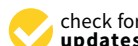

Physics and Astronomy Department, University of Tennessee, University of Tennessee Space Institute, Center for Laser Applications, 411 B.H. Goethert Pkwy, Tullahoma, TN 37388, USA; cparigge@tennessee.edu; Tel.: +1-931-841-5690

**Abstract:** This work communicates laser-plasma experiments in a gaseous mixture of hydrogen and nitrogen. Time-resolved spectroscopy measures the first four Balmer-series hydrogen lines together with selected neutral and ionized nitrogen lines. Optical breakdown plasma is generated in a 1:1 hydrogen:nitrogen mixture at ambient temperature and 0.27-atm pressure. Time-resolved spectroscopy records emitted radiation with spatial resolution along the slit height for the $H_\alpha$, $H_\beta$, $H_\gamma$, and $H_\delta$ lines. For 13 selected time delays from 0.25 µs to 3.25 µs and 0.025 µs gate-widths, micro-plasma diagnostics is evaluated. Of interest are the peak separation and width of $H_\delta$ and width of $H_\gamma$ for electron densities in the range of 0.1 to 1.0 $\times 10^{17} cm^{-3}$, and comparisons with $H_\beta$ and $H_\alpha$ diagnostics. Integral inversions interrogate spatial distributions of the plasma expansion. Applications include laboratory and stellar astrophysics plasma diagnosis.

**Keywords:** atomic and molecular spectroscopy; time-resolved spectroscopy; laser-plasma; laser-induced optical breakdown; stellar astrophysics spectra; white dwarf stars; hydrogen

## 1. Introduction

Measurement of laser-plasma has gained significant attention in recent years [1,2]. Applications include determination of chemical composition of materials, but always in view of accurate descriptions from a thermodynamic point of view [3,4]. However, addition of fractionally small amounts of hydrogen [5] can be vital for analysis of plasma. Time-resolved plasma spectroscopy [6,7] requires spatial resolution for appropriate characterization of expansion phenomena and species distributions. Analysis of line-of-sight data frequently employs integral inversions [8] for detailed interpretation.

Recent hydrogen laser-plasma experiments address the first four Balmer-series hydrogen lines—early in the plasma decay, neutral and ionized nitrogen lines are identified and appear first as the plasma cools. Plasma characteristics for time delays up to 0.275 µs were previously discussed for exclusively hydrogen gas at 0.75 atm and ambient temperature [9–11]. In this work, optical breakdown plasma is generated in a 1:1 hydrogen:nitrogen mixture at ambient temperature and 0.27-atm pressure. Plasma diagnosis is applied for time delays in the range of 0.25 µs to 3.25 µs [12,13]. Nitrogen contributes to the breakdown plasma seven times more electrons than hydrogen in cases of full ionization. Time-resolved spectroscopy records emitted radiation with spatial radiation along the slit height for the $H_\alpha$, $H_\beta$, $H_\gamma$, and $H_\delta$ lines.

Applications of the reported work include analysis of stellar astrophysical spectra from white dwarf stars [13]. In the visible region of the electromagnetic spectrum, $H_\beta$ is of primary interest in the characterization of white dwarf (WD) stars. However, comparisons of hydrogen Balmer-series emission lines with recorded astrophysical WD absorption lines requires bound-free opacity corrections [14]. The temperature of the closest WD to earth, Sirius B ($\alpha$ CMa B), is of the order of 30 kK [15] . Sirius B accompanies the brightest star Sirius A, as seen from earth. However, challenges in comparisons of

laboratory and astrophysical plasma-spectra have been reviewed recently [16], i.e., comparisons of $H_\beta$ line shapes of micro-plasma recorded with time-resolved emission spectroscopy and of $H_\beta$ line shapes in astrophysical white dwarf macro-plasma absorption spectra that are measured continuously at various observatories.

## 2. Materials and Methods

Laboratory laser-plasma measurements employ a pulsed, Q-switched, Nd:YAG laser device (Q-Smart 850 Quantel laser, USA) operated at a pulse-width of 6 ns and a pulse energy of 850 mJ at the wavelength of 1064 nm. Laser-induced optical breakdown is generated by focusing 150 mJ per pulse of ir fundamental radiation to achieve an irradiance of the order of 1 TW/cm$^2$ in a cell containing a 1:1 mixture of hydrogen and nitrogen, introduced at a pressure of 0.135 atm each after establishing a nominal mercury-diffusion-pump vacuum in the cell of the order $10^{-5}$ mbar. A crossed Czerny–Turner spectrometer (Jobin Yvon 0.64 m triple spectrometer, France) of 0.64-nm focal length disperses the emission spectra. The pulsed radiation is focused into the cell with the beam propagating from the top and parallel to the vertical 100 μm spectrometer slit. Further details of the experimental arrangement that is similar to the ultra-pure hydrogen experiments were communicated previously [9–11].

The spectral resolution amounts to 0.1 nm for the selected 1200 g mm$^{-1}$ holographic grating following corrections of the wavelength variation along the slit direction [12]. Of the order of 24-nm spectral coverage for the 1024 pixels along the wavelength-dimension, the 0.1-nm resolution corresponds to an instrument-prompt width of on-average 4.25 pixels. Grouping four pixels along the slit dimensions corresponds to a spatial resolution of 54.4 μm as the pixel area amounts to 13.6 μm × 13.6 μm.

## 3. Results

For 13 selected time delays from 0.25 μs to 3.25 μs and 0.025 μs gate-widths, micro-plasma data are captured. Of interest are the peak separation and width of $H_\delta$ and width of $H_\gamma$ for electron densities in the range of 0.1 to $1.0 \times 10^{17}$ cm$^{-3}$, and comparisons with $H_\beta$ and $H_\alpha$ diagnostics. Integral inversion interrogates the lateral spatial distribution of the recorded line-of-sight plasma expansion.

### 3.1. Line-of-Sight Measurements

A total of 52 two-dimensional data sets are recorded for the four $H_\alpha$, $H_\beta$, $H_\gamma$, and $H_\delta$ Balmer-series lines, including several nitrogen and ionized nitrogen lines especially for time delays in the range of 0.25–0.75 μs following plasma initiation. Figures 1–4 illustrate spectral radiance, pseudo-colored, individually scaled maps of the four hydrogen lines for time delays of 0.75, 1.5, 2.25, and 3.25 μs.

Features of Figure 1 include, (i) full-widths at half maximum increase from the $H_\alpha$ line to the $H_\delta$ line; (ii) the maximum decreases from $H_\alpha$ to $H_\delta$; (iii) occurrence of a neutral nitrogen line near 399 nm in the $H_\delta$ map above a UV-shaded background—the broad $H_\delta$ line will show a peak separation for larger time delays; (iv) peak separation of the $H_\beta$ line, or the occurrence of a dip—this dip is not due to self-absorption—prediction of the Stark effect on the $H_\beta$ line is part of the reason for E. Schrödinger [17] being awarded the Nobel Prize in Physics in 1933 [18], shared with P. Dirac.

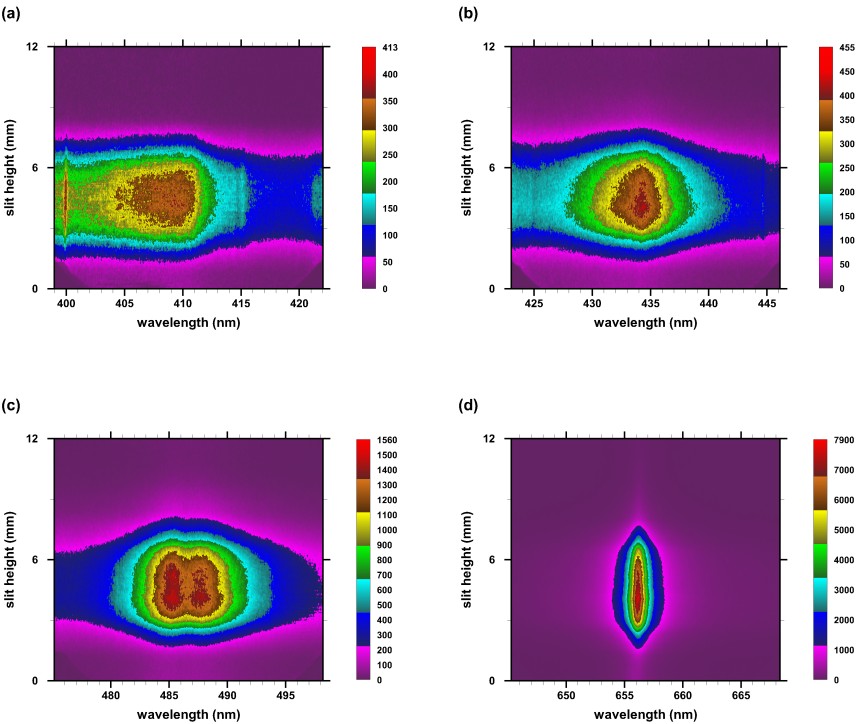

**Figure 1.** Recorded Balmer-series hydrogen lines at 0.75-μs time delay, 0.025-μs gate. (**a**) H$_\delta$, (**b**) H$_\gamma$, (**c**) H$_\beta$, and (**d**) H$_\alpha$.

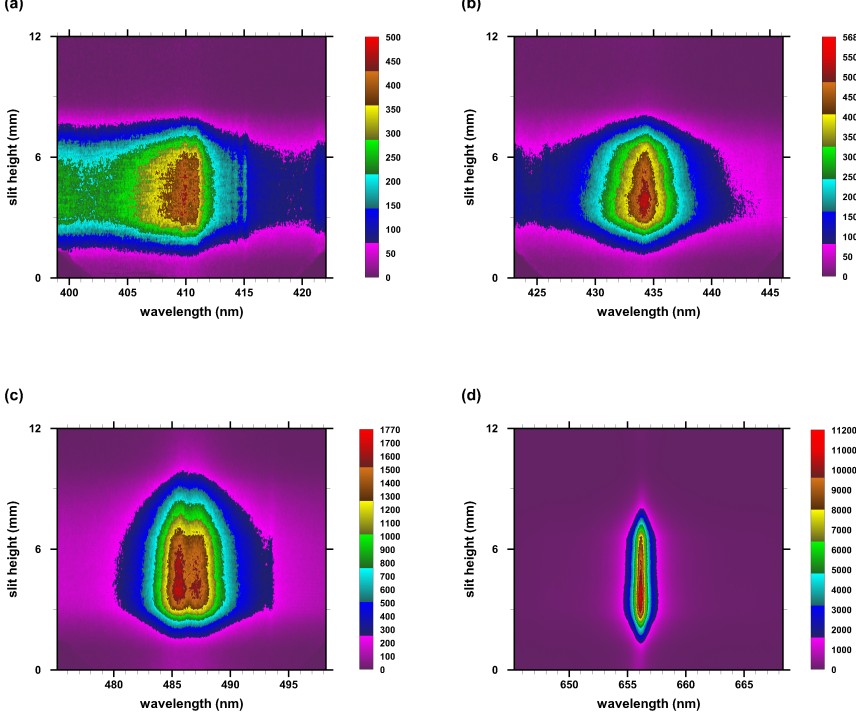

**Figure 2.** Recorded Balmer-series hydrogen lines at 1.5-μs time delay, 0.025-μs gate. (**a**) H$_\delta$, (**b**) H$_\gamma$, (**c**) H$_\beta$, and (**d**) H$_\alpha$.

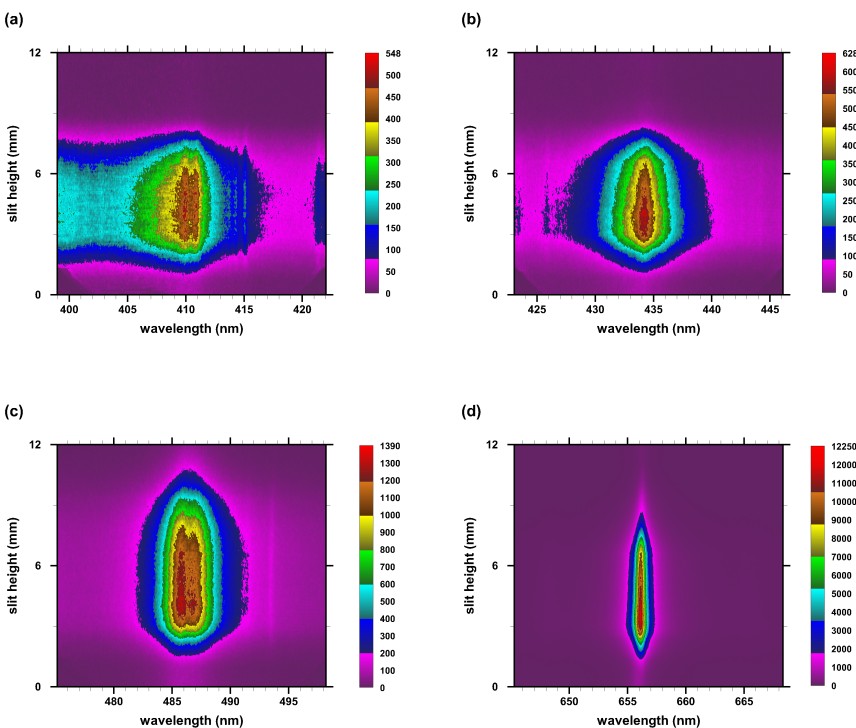

**Figure 3.** Recorded Balmer-series hydrogen lines at 2.25-μs time delay, 0.025-μs gate. (**a**) H$_\delta$, (**b**) H$_\gamma$, (**c**) H$_\beta$, and (**d**) H$_\alpha$.

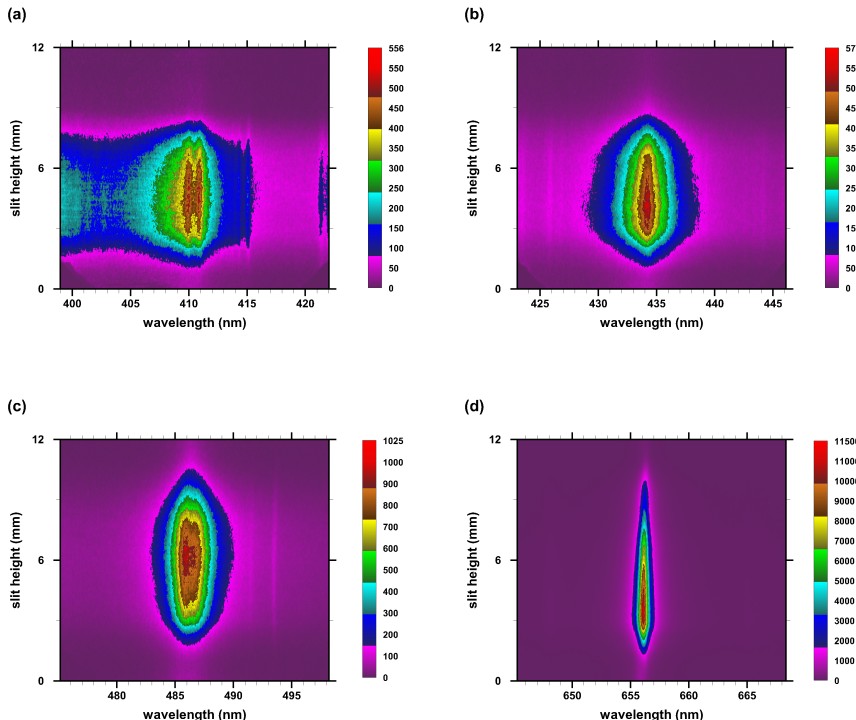

**Figure 4.** Recorded Balmer-series hydrogen lines at 3.25-μs time delay, 0.025-μs gate. (**a**) H$_\delta$, (**b**) H$_\gamma$, (**c**) H$_\beta$, and (**d**) H$_\alpha$.

In the images, detector dark-counts or background contributions are subtracted, sensitivity calibrations by reference to standard lamps are applied, and linear wavelength calibration is performed with penray lamps and by using calibrated spectrometer dials. The image of the spectrometer slit usually is slightly curved near the edges, consequently, wavelength calibrations are performed for each spectrum recorded along the slit. For the experiments reported here, four vertical pixels are combined to increase sensitivity requiring 256 individual wavelength calibrations for each of four spectrometer positions. The 256 recorded, wavelength calibrated Balmer-series spectra are slightly shifted and interpolated for display of the data versus slit height and wavelength position. In other words, the displayed maps are corrected for wavelength variations along the slit dimension.

Analysis of the laboratory emission spectra uses established empirical formulae for $H_\alpha$ and $H_\beta$ [9]. For $H_\alpha$, the width, $\Delta\lambda_\alpha$, and shift, $\delta\lambda_\alpha$, are indicators for electron density,

$$\Delta\lambda_\alpha[\text{nm}] = 1.3 \left( \frac{N_e[\text{cm}^{-3}]}{10^{17}} \right)^{0.64\pm0.03}, \tag{1}$$

$$\delta\lambda_\alpha[\text{nm}] = 0.055 \left( \frac{N_e[\text{cm}^{-3}]}{10^{17}} \right)^{0.97\pm0.03}. \tag{2}$$

Analysis of $H_\beta$ offers three indicators of electron density: Width, $\Delta\lambda_\beta$, peak separation, $\delta\lambda_{\beta-ps}$, and dip-shift, $\delta\lambda_{\beta-ds}$,

$$\Delta\lambda_\beta[\text{nm}] = 4.5 \left( \frac{N_e[\text{cm}^{-3}]}{10^{17}} \right)^{0.71\pm0.03}, \tag{3}$$

$$\delta\lambda_{\beta-ps}[\text{nm}] = 1.3 \left( \frac{N_e[\text{cm}^{-3}]}{10^{17}} \right)^{0.61\pm0.03}, \tag{4}$$

$$\delta\lambda_{\beta-ds}[\text{nm}] = 0.14 \left( \frac{N_e[\text{cm}^{-3}]}{10^{17}} \right)^{0.67\pm0.03}. \tag{5}$$

The $H_\beta$ dip-shift allows one to measure electron density [10] up to the $H_\beta$ Inglis-Teller limit [19] of $60 \times 10^{17} \text{cm}^{-3}$. However, $H_\beta$ is preferred in a variety of astrophysics data-reduction efforts for electron densities of the order of $10^{17} \text{cm}^{-3}$.

### 3.2. $H_\delta$ and $H_\gamma$ Line Profiles

The analysis of the $H_\delta$ and $H_\gamma$ data rely on computer simulations [20], and in this work, on published Stark tables [21] that only show electron-density data in the range of 0.1 to $1 \times 10^{17} \text{cm}^{-3}$ for a temperature of 20 kK. From the Stark tables, $H_\delta$ and $H_\gamma$ line-profiles can be constructed for electron densities, $N_e$, of 0.1 and $1 \times 10^{17} \text{cm}^{-3}$. Figures 5 and 6 illustrate the line shapes of $H_\delta$ and $H_\gamma$, respectively. Subsequently, $H_\delta$ and $H_\gamma$ full-width half maximum (FWHM) can be determined for plasma diagnosis, moreover, $H_\delta$ invites the use of peak separation (PS) as a diagnostic tool. The recorded data display expected Stark-effect trends [21–24].

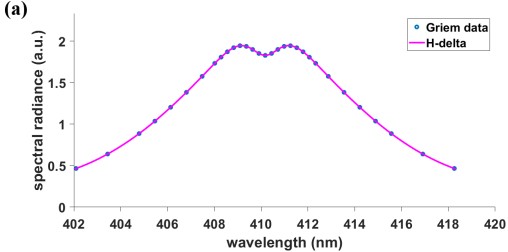 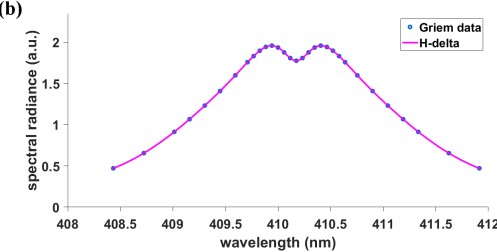

**Figure 5.** Line shapes of Balmer $H_\delta$ at a temperature of 20 kK [21]. (**a**) $N_e = 1 \times 10^{17} \text{cm}^{-3}$, FWHM: 10 nm, PS: 2.0 nm. (**b**) $N_e = 0.1 \times 10^{17} \text{cm}^{-3}$, FWHM: 2.2 nm, PS: 0.5 nm.

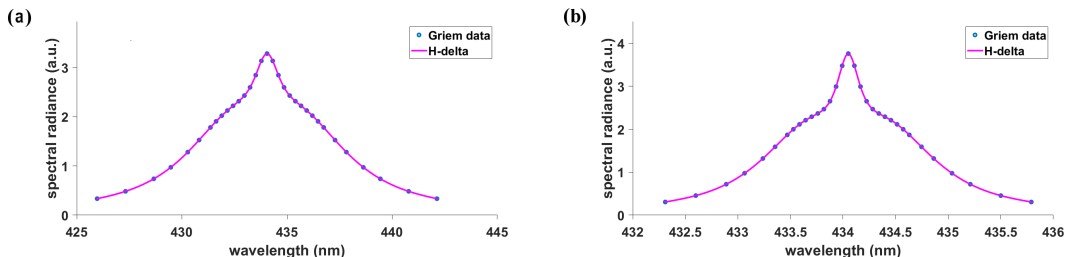

**Figure 6.** Line shapes of Balmer $H_\gamma$ at a temperature of 20 kK [21]. (**a**) $N_e = 1 \times 10^{17}$cm$^{-3}$, FWHM: 6.0 nm. (**b**) $N_e = 0.1 \times 10^{17}$cm$^{-3}$, FWHM: 1.15 nm.

From log-log fitting of $H_\gamma$ and $H_\delta$ tables, see Figures 5 and 6, in the range of 0.1–1 $\times 10^{17}$cm$^{-3}$, one obtains for $H_\delta$ FWHM, $\Delta\lambda_\delta$,

$$\Delta\lambda_\delta[\text{nm}] = 10 \left( \frac{N_e[\text{cm}^{-3}]}{10^{17}} \right)^{0.67}, \tag{6}$$

and the $H_\delta$ peak separation, $\Delta\lambda_{\delta-\text{ps}}$, in the range of 0.1–1. $\times 10^{17}$cm$^{-3}$ amounts to

$$\Delta\lambda_{\delta-\text{ps}}[\text{nm}] = 2.0 \left( \frac{N_e[\text{cm}^{-3}]}{10^{17}} \right)^{0.62}. \tag{7}$$

The $H_\delta$ shows peak separations just like those for the $H_\beta$ line [25]. The appearance of the $H_\gamma$ line is perhaps unusual in view of the $H_\alpha$ line, but Figure 6 portrays the line shape modifications due to tabulated data [21] describing the Stark effect. From log-log fitting, one obtains for $H_\gamma$ FWHM, $\Delta\lambda_\gamma$,

$$\Delta\lambda_\gamma[\text{nm}] = 6.0 \left( \frac{N_e[\text{cm}^{-3}]}{10^{17}} \right)^{0.72}. \tag{8}$$

Figure 7 portrays examples of average $H_\delta$ and $H_\gamma$ line shapes for the 1:1 hydrogen:nitrogen mixtures at a time delay of 1.5 µs. The average is taken over the entire slit dimension. The $H_\delta$ line shows the expected double-peak appearance, and the $H_\gamma$ line barely indicates the line shape illustrated in Figure 6 and predicted in the Stark tables [21]. Further detailed comparisons should begin with pure hydrogen experiments and avoid averaging along the slit dimension. However, analysis of line-of-sight data along the slit height is further discussed in Section 3.5.

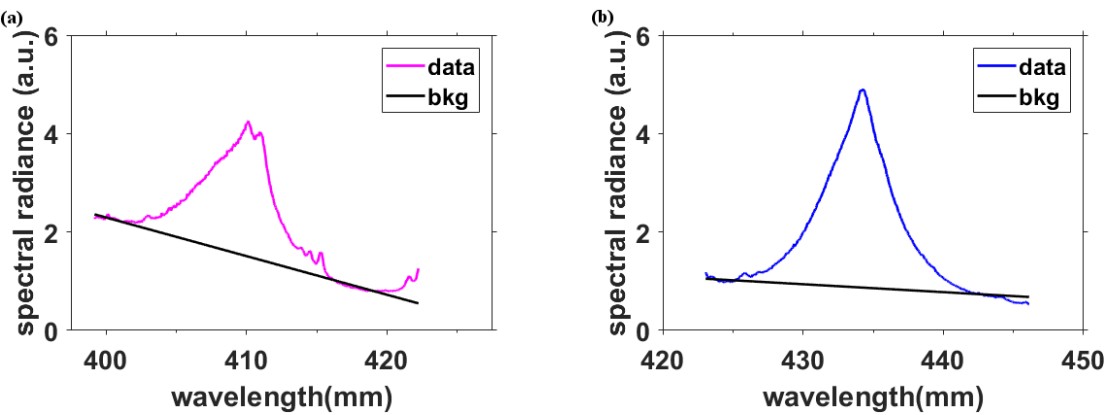

**Figure 7.** Measured line shapes at a time delay of 1.5 µs. (**a**) $H_\delta$ and (**b**) $H_\gamma$.

### 3.3. Abel-Inverted Spectral Maps

The recorded line-of-sight data are further processed using inverse Abel transform to examine the spatial distribution of the expanding plasma. The wavelength variation along the slit dimension is corrected for each spectral line of a single slit height versus wavelength map. Consequently, the maps in Figures 1–4 are primed for analysis of integral inversion of the recorded line-of-sight data. The central spectral regions appear almost symmetrical with-respect-to the slit height. The analysis method includes determination of the central line along the slit dimension, followed by generation of the mirror image for Abel-inversion. The top-bottom asymmetry is then evaluated analogous to a previously communicated approach [26] that shows variation in electron density but largely within error bars. This work only displays Abel-inverted images without correction for the asymmetry in the recorded data. Yet the inferred electron temperature along the slit exhibits variation indicative of fluid-physics phenomena that are expected for laser-plasma shock-wave expansion, as further discussed in Section 3.5. For the 3.25-μs time delay, only slight deviations are anticipated due to diminished shockwave effects. Figures 8 and 9 illustrate the spectral radiance as function of radius and wavelength for time delays of 1.5 and 3.25 μs, respectively.

The Abel-inverted, 3.25 μs time-delay data display reasonably smooth line shapes for all four Balmer-series lines. The $H_\delta$ peak separation can be easily demarcated, but the $H_\beta$ peak separation is difficult to extract. In other words, The $H_\delta$ line provides both peak separation and FWHM diagnostics as the electron density decreases in the range of 1.0 to 0.1 $\times 10^{17} cm^{-3}$. A 1-nm peak separation of the $H_\delta$ line in Figure 8a implies $N_e = 0.33 \times 10^{17} cm^{-3}$.

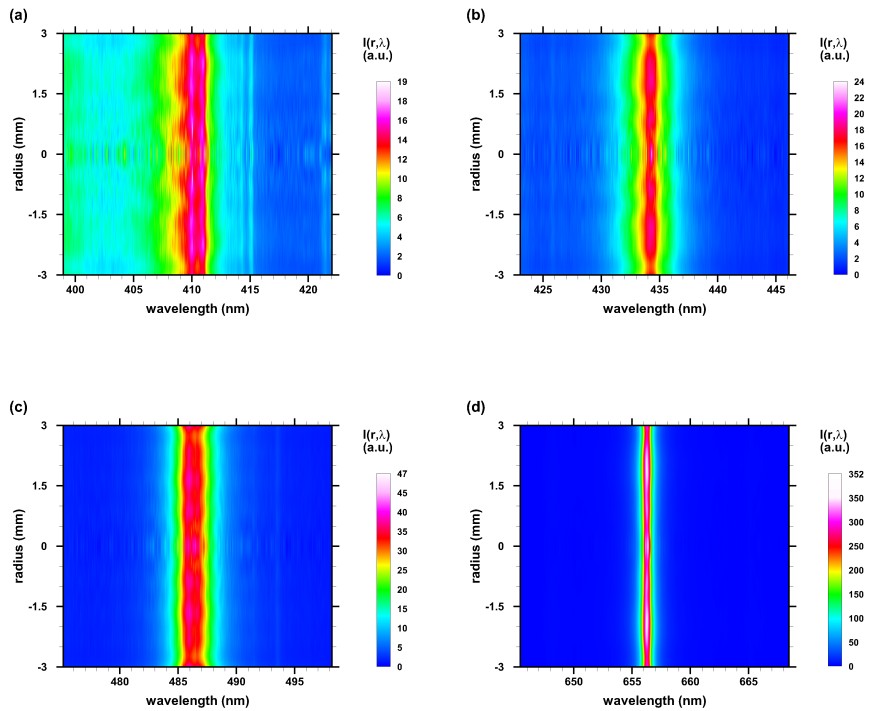

**Figure 8.** Abel-inverted Balmer-series hydrogen lines at 3.25-μs time delay, 0.025-μs gate. (**a**) $H_\delta$, (**b**) $H_\gamma$, (**c**) $H_\beta$, and (**d**) $H_\alpha$.

For the 1.5 μs data, the Abel-inverted, first four Balmer-series lines reveal minima at the center of the plasma. These minima are representative of plasma expansion phenomena following initiation of optical breakdown. Figure 9a,c reveal peak separation of the $H_\delta$ and $H_\beta$ lines, respectively. The $H_\delta$ map also indicates a background shaded towards the UV, and there appear to be contributions from

the H$_\epsilon$ line (at the low colorgreen wavelength side in the map). The H$_\delta$ line shows a shallower dip for higher electron density as illustrated in the range of 0.1 to 1 $\times 10^{17}$cm$^{-3}$ for Figure 5. However, the H$_\beta$ peak separation indicates an electron density of $1.5 \times 10^{17}$cm$^{-3}$.

Both Figures 8 and 9 indicate undulations of the signal along the radial direction. Such undulations can be expected as the laser-plasma expands—in related experiments of optical breakdown in air, multiple reflections can be seen in shadowgraphs that are captured following optical breakdown in air [27]. Similar behavior is expected to occur for the hydrogen:nitrogen mixture. Figure 10 displays recorded air-breakdown shadowgraph to further elucidate multiple reflections that are likely causing undulations in the plasma core. The shadowgraph experiments recorded 125 images for each time delay, the selected images portray typical images.

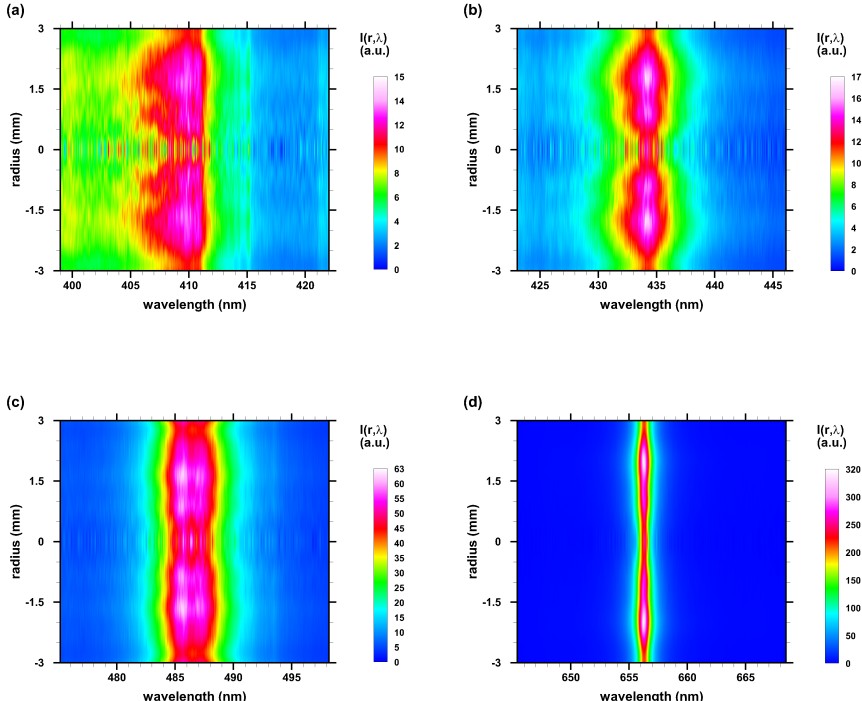

**Figure 9.** Abel-inverted Balmer-series hydrogen lines at 1.5-μs time delay, 0.025-μs gate. (**a**) H$_\delta$, (**b**) H$_\gamma$, (**c**) H$_\beta$, and (**d**) H$_\alpha$.

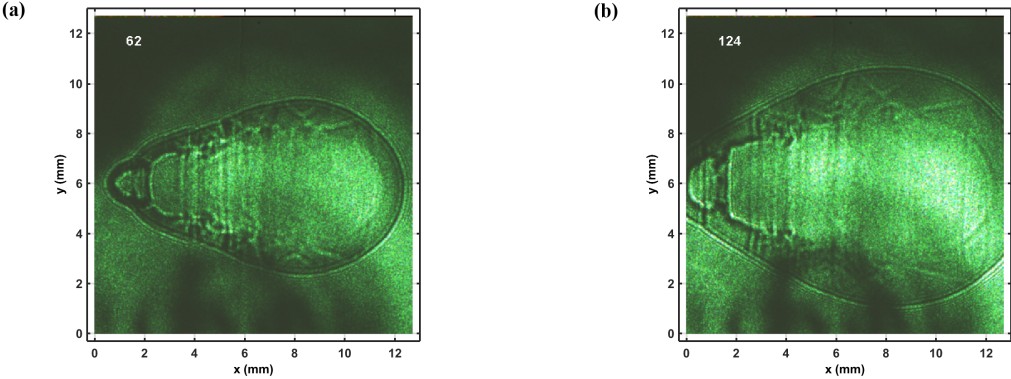

**Figure 10.** Shadowgraphs of optical breakdown in laboratory air. The laser beam propagates from the right, images are captured using a second, imaging laser. The "bubble" on the laser side indicates absorption of laser radiation. Time delay: (**a**) 1.5 μs, image 62. (**b**) 3 μs, image 124.

### 3.4. Electron Temperature and Density

Analysis of the recorded spectra along the slit height aims to evaluate electron temperature, $T_e$, from Boltzmann plots. Moreover, electron density, $N_e$, determination relies on formulae and tables that describe Stark broadening in laser-plasma.

Figure 11 portrays typical Boltzmann plots from line-of-sight measurements at time delays of 2.25 and 3.25 µs. The method uses Boltzmann distributions and determination of the slope to extract the temperature from the negative of 1/slope. The intercept of the straight line is not used for evaluation of the temperature. For data points from the four Balmer-series members close to the fitted line, one can infer local thermodynamic equilibrium. As the data points deviate from the fitted line, deviations from equilibrium may be concluded—self-absorbed lines would also cause deviations of data points from the straight line, and equally, ambiguities in determination of the baseline. However, expansion dynamics may lead to variations in the integrated line intensities leading to a 25% estimate for the accuracy of the inferred temperature.

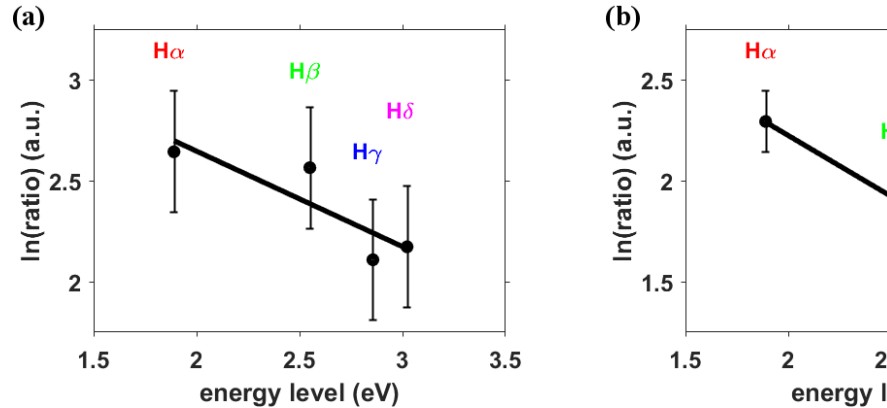

**Figure 11.** Boltzmann plot from line-of-sight data at 4.8-mm slit height. (**a**) T = 26.5 kK, time delay 1.5 µs (Figure 2). (**b**) T = 19.3 kK, time delay 3.25 µs (Figure 4).

The electron density is related to the hydrogen Balmer-series FWHM to the power of the order of 1.5; however, detailed theory predictions and experimental results indicate slight deviations. Empirical equations are obtained from log-log fitting of FWHM and electron density. For red-shift of the $H_\alpha$ line, the dependency on electron density is almost linear. Significant new results also use peak separation of the $H_\beta$ line to find $N_e$. Stark broadening widths increase from $H_\alpha$ to $H_\gamma$. The use of $H_\beta$ peak separation for $N_e$ determination in laser-plasma adds a new diagnostic tool. Investigations of the $H_\delta$ peak separation is communicated for $N_e$ in the range of 0.1 to $1 \times 10^{17}$ cm$^{-3}$ and for the hydrogen–nitrogen laser-plasma. Table 1 displays measured FWHM and peak separations. Table 2 reports the corresponding electron densities. The values are determined from the spectra at a slit height of 4.8 mm by using Matlab® software 'peakfit.m' [28]. Prior to determining the FWHM with 'peakfit.m', the different background contributions for the four hydrogen lines are estimated from the 50-point average of the recorded signals at the low- and high- wavelength regions for each of the four wavelength-windows. A straight line connecting the two averages serves as the wavelength-dependent background. Fitting of the four background-corrected data uses single Lorentzian line shapes, yet initially Voigt- line shapes and double-peaked Lorentzian line shapes were also investigated. For automated analysis, 'peakfit.m' screen outputs are suppressed, FWHM- and area-data provide measures of the electron density (see Equations (2)–(7)) and temperature, respectively. An analogous fitting approach accomplishes determination of the areas of the four lines, but corrections are applied for the area due to the recording of only the central portion of the lines. The analysis with Boltzmann plots uses hydrogen data quoted recently [11]. While the wings of hydrogen lines may not be of Lorentzian shape, this work relies on Lorentzian line shapes for estimates of the area corrections

that are give-or-take 10%. Other software packages for display of measured spectra include Tecplot®
and Matlab®.

**Table 1.** First four Balmer-series FWHM and $H_\delta$ and $H_\beta$ peak separations from line-of-sight data at
4.8-mm slit height.

| $\tau$ [µs] | $\Delta\lambda_\delta$ [nm] | $\Delta\lambda_{\delta-ps}$ [nm] | $\Delta\lambda_\gamma$ [nm] | $\Delta\lambda_\beta$ [nm] | $\Delta\lambda_{\beta-ps}$ [nm] | $\Delta\lambda_\alpha$ [nm] |
|---|---|---|---|---|---|---|
| 1.50 | $7.5 \pm 1.5$ | $1.4 \pm 0.8$ | $5.0 \pm 0.2$ | $4.2 \pm 0.25$ | $1.2 \pm 0.2$ | $1.17 \pm 0.1$ |
| 3.25 | $5.8 \pm 0.3$ | $1.1 \pm 0.2$ | $3.2 \pm 0.2$ | $2.6 \pm 0.25$ | $0.8 \pm 0.15$ | $0.78 \pm 0.1$ |

**Table 2.** First four Balmer-series electron densities, $N_e$ $[10^{17} cm^{-3}]$, from line-of-sight data at 4.8-mm
slit height.

| $\tau$ [µs] | $N_e(\delta)$ | $N_e(\delta - ps)$ | $N_e(\gamma)$ | $N_e(\beta)$ | $N_e(\beta - ps)$ | $N_e(\alpha)$ | **Average $N_e$** |
|---|---|---|---|---|---|---|---|
| 1.50 | 0.65 | 0.56 | 0.77 | 0.91 | 0.88 | 0.84 | **0.77** |
| 3.25 | 0.44 | 0.38 | 0.41 | 0.46 | 0.46 | 0.45 | **0.43** |

For the 3.25-µs time delay, electron densities largely agree when inferred from the widths of the
four Balmer-series lines and from the peak separations of $H_\beta$ and $H_\delta$. However, there are subtle
differences of $N_e$ determined at 1.5 µs time delay. One may conclude from the appearance of the
measured emission spectra and the computed Abel-inverted spectra that super- to hyper- sonic
laser-plasma expansion speeds affect the establishment of local thermodynamic equilibrium. Details
are investigated in the next section.

### 3.5. Plasma Expansion Dynamics

Laser-plasma generation with laser pulses of the order of 10 ns and pulse energies of 80 to 800 mJ
ambient laboratory conditions causes a shockwave that expands at a rate of one to a few mm per µs,
or a few km/s, for time delays of the order of 1 µs. Early in the plasma decay, expansion speeds of up
to 80 km/s are usually encountered that may be described in engineering terms as well above re-entry
speeds or high hypersonic speeds. Several tens of microseconds after plasma initiation, the shockwave
reduces to the speed of sound.

In view of the "bubble" expanding in air and the associated isentropic expansion, one can
determine electron density or temperature to elucidate the phenomena. Captured shadowgraphs
can guide time-resolved spectroscopy; however, there should be an indication of the shockwave in
spatially and temporally resolved measured spectra. One approach may use Radon- or Abel- inversion
techniques [8] that are generally known as computed tomography methods. Or one may closely
investigate the captured spectra along the slit height.

Systematic determination of the Stark widths from line-of-sight data is expected to reveal a higher
electron density near the shockwave than in the plasma core. Alternatively, determination of the
area of the atomic Balmer-series lines should indicate higher temperature near the shockwave than in
the plasma center. Moreover, for time delays of several µs—as the shockwave expanded beyond the
interrogated volume—the plasma is expected to be homogeneous and indicate local thermodynamic
equilibrium for electron densities in excess of $10^{16}$ cm$^{-3}$, as indicated by the necessary McWhirter
criterion [3,4], but with higher temperature in the core.

Figures 12 compares electron-density results for a time delay of 1.5 µs. Increases in electron
density at the plasma edges are indications of the expanding shockwave. In order to estimate the
shockwave radius as function of time delay, $\tau$, it is advantageous to use the Taylor–Sedov formula [29]
for spherically expanding plasma,

$$R(\tau) = \left(E_p/\rho\tau^2\right)^{1/5}, \tag{9}$$

where the laser-pulse energy, $E_p$, for the experiments amounts to 0.15 J, and the density of the 1:1
nitrogen hydrogen mixture, $\rho$, equals 0.37 kg/m$^3$. For $\tau = 1.5$ µs, one finds for the radius R = 4 mm.

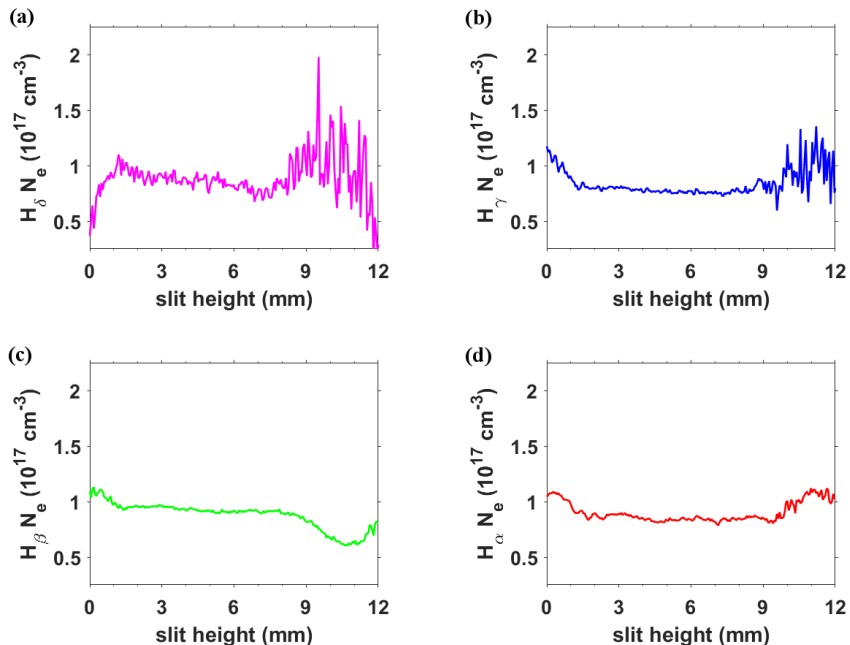

**Figure 12.** Electron density from line-of-sight data displayed in Figure 2 at $\tau = 1.5$ μs, 0.025-μs gate. (**a**) $H_\delta$, (**b**) $H_\gamma$, (**c**) $H_\beta$, and (**d**) $H_\alpha$.

The electron-density variation along the slit dimension indicates that the electron density determined from $H_\beta$ is smaller than obtained from $H_\alpha$ for slit heights larger than 9 mm. This result may be associated with variations induced along the shockwave front. Further investigation is aimed at determining the electron temperature variation. Figure 13 illustrates spectroscopic snapshots of the determined temperature for $\tau = 1.5$ μs and $\tau = 2.5$ μs. The snapshots, computed from Boltzmann plots and for equally weighted contributions from all four lines, illustrate that noticeable expansion occurs towards the incoming laser beam and propagates towards the top of the slit height near 12 mm in Figure 13b.

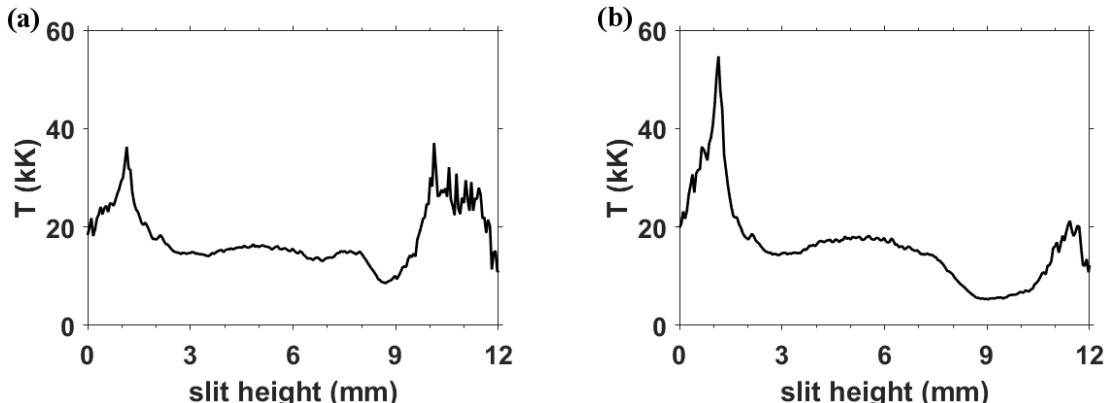

**Figure 13.** Electron temperature from line-of-sight data. (**a**) $\tau = 1.5$ μs, 0.025-μs gate. Average temperature: 17.9 kK. (**b**) $\tau = 2.5$ μs, 0.025-μs gate. Average temperature: 16.3 kK.

## 4. Discussion

The experimental study of $H_\gamma$ and $H_\delta$ line shapes shows that results are obtained that are consistent with those from $H_\alpha$ and $H_\beta$, especially for time delays of the order of a few μs after initiation of laser-plasma: Shock waves tend to propagate at speeds of the order of one mm/μs for the 0.27-atm hydrogen–nitrogen mixture. Investigations of time-resolved spectra for time delays of the order of 0.25

to 1.5 μs reveal effects from the expanding shockwave. Analysis of spectra that are recorded along the slit dimension indicate a temperature profile that is expected from computations of shockwave expansions. Abel-inverted spectra confirm the shockwave increase of temperature and electron density, including a reduced temperature and density profile in the center of the expanding micro-plasma.

The explored $H_\gamma$ and $H_\delta$ lines can provide additional diagnostics laser-plasma and for analysis of white dwarf spectra, especially in the range of 0.1 to $1 \times 10^{17} \text{cm}^{-3}$. However, when using all four lines of the Balmer-series, bound-free opacity corrections would be required for comparisons of laboratory emission with white dwarf absorption spectra. We conclude, analysis of white-dwarfs and other astrophysical objects such as active galactic nuclei from only one line, i.e., $H_\beta$, may be preferred as bound-free effects may not be significant across one line for purposes of determinations of the $H_\beta$ width and asymmetry. Future work however should include new results from laboratory measurements that address specifically the shapes of lines for diagnosis of astrophysics phenomena.

**Funding:** The authors appreciate the support in part by the Center for Laser Application, a State of Tennessee funded Accomplished Center of Excellence at the University of Tennessee Space Institute.

**Conflicts of Interest:** The author declares no conflict of interest.

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
