# Peer review of "Measurements of Gaseous Hydrogen–Nitrogen Laser-Plasma"

_atoms, doi:10.3390/atoms7030061_

Round 1
Reviewer 1 Report
Manuscript ID: atoms-537937
Title: Measurements of gaseous hydrogen-nitrogen laser-plasma
Authors: Christian Parigger *
Sp.issue:Laser Plasma Spectroscopy Applications
https://www.mdpi.com/journal/atoms/special_issues/LaserPlasmaSpec_App
In this paper author present results of laboratory laser- plasmas research. The paper contains a lot of very good experimental work, backed up where needed with calculations. The results are important for the laboratory research and other scientific applications. I would like to recommend the publication but with few (mainly) technical requests.
Some requests:
In affiliation: Author Orcid ID is wrong. Please give correct one
https://orcid.org/0000-0003-1286-7405
line 33: Please insert proper Refs. which supports info. about Sirius B plasma conditions
e.g.
\bibitem[Gianninas et al.(2011)]{2011ApJ...743..138G} Gianninas, A., Bergeron, P., \& Ruiz, M.~T.\ 2011, \apj, 743, 138 or similar
Line 148. I would suggest that the author extend part about curve fitting tool and Matlab® software “peakfit.m" with one or two sentences with details and also about other software they use in current work.
Page 11 Line 192: Please give Ref. which support sentence “Analysis of white-dwarfs and other astrophysical objects such as active galactic nuclei … widths and asymmetries of the H_{\beta} line.” or say “We conclude …”
Typos:
Page 6, text before Eq. 8: modifcations => modifications
Page 6, line 112: wavelenght => wavelength
Page 9, line 146: rang =>range
Page 10, text before Eq. 9: speherically => spherically
Page 10 line 178: sepctroscopic => spectroscopic
Respectfully,
Author Response
Dear reviewer,
I appreciate your comments. Thank you! The edits are indicated in green. (Other edits in blue.)
(1) Yes, the "orcid" needed to be included in the latex manuscript file;
(2) Included new Reference [15];
(3) Added description of software use, lines 149 to 161;
(4) Inserted "We conclude... " as suggested, see line 204;
(5) corrected typos in or near lines 101, 112, 146, 187, 190.
Respectfully,
Reviewer 2 Report
The MS presents measurements of the first four transitions of the Balmer series recorded in optical breakdown plasma experiment. The line-shapes and relative intensities are used to derive the electron density and temperature. A short discussion is given on the dynamics of the laser-produced plasma in view of the spatially- and temporally-resolved spectroscopic results.
While I find the results to be interesting, there are a few issues that require clarifications in the text before publication.
1. There is no description of Figs. 5 and 6 in the text. Consequently, it is difficult to understand their purpose. Are they just for demonstrating Griem’s data? Also, the legends of the Figures are confusing. What are the curves in these figures?
2. Section 3.3. Abel-inverted spectral maps: The spectra in Figs 3-4 are unsymmetrical. How these asymmetries were treated? Do they lead to large uncertainties in the inferred density distribution?
3. The line width and peak-separation are used for the electron density determination. However, it is also very interesting to compare the overall recorded line shape with theory, particularly in light of the comment on the H_gamma shape “The appearance of the H_gamma line is perhaps unusual” (see text after Eq. 7). At least one or two examples of such comparisons should be given in the MS.
4. Fig. 11 – The author should comment on the apparent decreasing density for h > 9 mm indicated by the H_b, whereas the other lines seem to indicate the opposite.
Style and typos:
Line 6: – “s” is missing
Line 43: – “ to achieve … of the order ...“
Lines 81 – 86: - The same information is already given before.
Author Response
Dear Reviewer,
Your comments are greatly appreciated. Edits are indicated in blue (other edits are in green)
(1) I included a sentence regarding Figures 5 and 6, new line 92, and edited the captions;
(2) New lines 96 to 101 and new Figure 7 discuss line shapes of H-delta and H-gamma. The two examples in Figure 7 compare measured with predicted (from Griem's Stark tables) line profiles.
(3) New lines 107 to 114 expand on Abel-inversion, including addition of new Ref. [26];
(4) New lines 199 to 202 address the electron density variation of H-beta
(5) Corrected the typo in line 6, added a noun in line 43, and removed lines 81-86.
Respectfully,